# Policy Challenges in Ultra-Rare Cancers: Ethical, Social, and Legal Implications of Melanoma Prevention and Diagnosis in Children, Adolescents, and Young Adults

**DOI:** 10.3390/healthcare13030321

**Published:** 2025-02-04

**Authors:** Pietro Refolo, Costanza Raimondi, Livio Battaglia, Josep M. Borràs, Paula Closa, Alessandra Lo Scalzo, Marco Marchetti, Sonia Muñoz López, Joan Prades Perez, Laura Sampietro-Colom, Dario Sacchini

**Affiliations:** 1Department of Health Care Surveillance and Bioethics, Section of Bioethics and Medical Humanities, Università Cattolica del Sacro Cuore, 00168 Rome, Italy; costanza.raimondi1@unicatt.it; 2Research Centre for Clinical Bioethics & Medical Humanities, Università Cattolica del Sacro Cuore, 00168 Rome, Italy; 3National Agency for Regional Health Services (AGENAS), 00187 Rome, Italy; battaglia@agenas.it (L.B.); loscalzo@agenas.it (A.L.S.); marchetti@agenas.it (M.M.); 4Department of Clinical Sciences, University of Barcelona, 08036 Barcelona, Spain; jmborras@iconcologia.net; 5Fundació de Recerca Clínic Barcelona, Institut d’Investigacions Biomèdiques August Pi i Sunyer, 08036 Barcelona, Spain; munoz6@recerca.clinic.cat (S.M.L.); lsampiet@clinic.cat (L.S.-C.); 6Health Department, University of Barcelona, 08036 Barcelona, Spain; jprades@iconcologia.net

**Keywords:** ultra-rare cancer, childhood melanoma (CHM), artificial intelligence (AI), prevention, early diagnosis, ethical, legal and social implications (ELSIs)

## Abstract

**Background:** The ultra-rare nature of melanoma in children, adolescents, and young adults poses significant challenges to the development and implementation of effective prevention and diagnostic strategies. This article delves into the ELSIs surrounding these strategies, placing particular emphasis on the transformative potential of AI-driven tools and applications. **Methods:** Using an exploratory sequential mixed methods approach, this study integrated a PICO-guided literature review and qualitative insights from two focus groups. The review included 26 peer-reviewed articles published in English from January 2019 to January 2024, addressing ELSIs in melanoma, rare diseases, and AI in dermatology. Focus groups included a March 2024 session in Berlin with 15 stakeholders (patients, caregivers, advocates, healthcare professionals) and a November 2024 online session with 5 interdisciplinary experts. **Results:** Six key priorities for healthcare policies emerged: addressing cultural factors, such as the glorification of tanned skin; enhancing professional training for accurate diagnosis; balancing the risks of overdiagnosis and underdiagnosis; promoting patient autonomy through transparent communication; reducing inequalities to ensure equitable access to care; and making ethical and legal use of AI in healthcare. **Conclusion:** These priorities provide a comprehensive framework for advancing the prevention and diagnosis of melanoma in children, adolescents, and young adults, leveraging AI technologies while prioritizing equitable and patient-centered healthcare delivery.

## 1. Introduction

Skin melanoma is the fifth most commonly diagnosed cancer in men (after prostate, lung, colorectal, and bladder cancers) and in women (after breast, colorectal, lung, and corpus uteri cancers) in EU-27 countries, with 106,369 new cases reported in 2020 (50,972 in men and 55,397 in women) [1]. Despite prevention efforts, its global prevalence and associated disability-adjusted life years (DALYs) have been increasing since 1990 [2].

Melanoma in children, adolescents, and young adults (henceforth referred to as “childhood melanoma”, CHM) occurs in individuals 20 years old or younger. It is an ultra-rare malignancy, with 1.3–1.6 cases per million in children under 15 and 15 cases per million in adolescents aged 15–19. Its incidence has been rising by 4.1% annually in adolescents since 1997 [3]. Despite its rarity, CHM remains the most common malignant skin cancer in this age group in EU-27 countries [4].

Similar to other rare diseases, rare cancers are characterized by healthcare inequities, insufficient information, limited research opportunities, diagnostic difficulties, uncoordinated care, and inadequate prevention policies [5]. These inequalities are especially pronounced in pediatric groups, where fatality rates for uncommon cancers remain elevated compared to those of more prevalent cancers, even in the face of substantial progress in therapeutic approaches [5].

Melanoma prevention and clinical management aim to reduce mortality and morbidity by lowering incidence, enhancing early detection, delaying disability onset, and improving quality of life [6]. Prevention includes the following:(1)Primary prevention: reducing ultraviolet (UV) exposure through physical barriers, topical agents, or systemic protection;(2)Early detection and intervention through screening, dermoscopy, and imaging. More recently, artificial intelligence (AI)-driven tools, complemented by professional examinations and self-monitoring, have broadened the array of available methods for melanoma detection;(3)Tertiary prevention: mitigating disease impact in diagnosed individuals to reduce complications and improve outcomes.

The ultra-rare nature of CHM presents significant challenges in designing and implementing effective prevention strategies.

This article explores the ethical, legal, and social implications (ELSIs) related to the prevention and diagnosis of CHM, with a particular focus on the emerging role of AI-driven tools and applications, which have the potential to revolutionize early detection. In fact, dermatology is one of the fields where AI has been most extensively utilized, offering innovative solutions for skin cancer detection, diagnosis, and management. Its ability to analyze large datasets, identify subtle patterns, and provide accurate predictions has made AI an invaluable asset in enhancing diagnostic precision and improving patient outcomes.

This paper is part of the European Union project “Novel Health Care Strategies for Melanoma in Children, Adolescents, and Young Adults” (MELCAYA) (https://www.melcaya.eu/, accessed on 26 December 2024), funded by the European Union (grant agreement ID 101096667). The project seeks to deepen the understanding of melanoma risk factors and determinants to enhance prevention, diagnosis, and prognosis for melanoma in childhood melanoma. Within this framework, our Work Package focuses specifically on developing tools and guidance for health authorities to support decision-making processes related to the early diagnosis and prevention of melanoma in children.

In line with the specific objectives of the MELCAYA project, the research question has been refined into three sub-questions: A. What are the ELSIs associated with prevention and diagnosis of CHM? B. What are the ELSIs associated with prevention and diagnosis of (ultra-)rare diseases? C. What are the ELSIs associated with the use of AI for prevention and diagnosis within the broader field of dermatology? By addressing these critical considerations, this article aims to provide insights and guidance to inform the development of comprehensive and effective prevention strategies tailored to the unique challenges posed by CHM.

## 2. Materials and Methods

This study employed an exploratory sequential mixed methods design [7,8], beginning with a literature review followed by two qualitative focus groups to comprehensively address the research questions. This approach ensured that both the evidence from the literature and the lived experiences of stakeholders were adequately represented. By integrating evidence from the literature with qualitative data, this approach ensured a holistic representation of both theoretical and practical viewpoints. It also enabled the identification of nuanced challenges and potential solutions that might not have emerged from the literature alone. The study was conducted in accordance with the step-by-step guide outlined by Skamagki et al. [9].

Literature Review: Three search strategies were developed utilizing a structured PICO (Population, Intervention, Comparator, Outcome) model to ensure both precision and alignment with the research objectives. The search was conducted in PubMed, focusing exclusively on peer-reviewed articles published in English from January 2019 to January 2024. Inclusion and exclusion criteria were established prior to the study and they are detailed in the Appendix A.

Two investigators (PR and CR) screened the studies using 2024 Rayyan (https://www.rayyan.ai/, accessed on 26 December 2024) independently examining the titles and abstracts of the documents retrieved from the search that met the inclusion criteria, and ambiguous cases were discussed with DS to reach consensus. Additionally, relevant publications identified through manual searches were included. Each selected study was read at least twice to ensure a thorough understanding before data extraction. Two reviewers (P.R. and C.R.) independently extracted data using a predefined Excel form, capturing details such as practices, participant types, general considerations, and ethical, legal, and social issues, and a consensus meeting was held to compare extractions and resolve discrepancies by jointly reviewing the original full text and agreeing on the correct information. The study selection and data extraction are detailed in the Appendix A.

Focus Groups: Two focus groups were conducted to gather qualitative insights from stakeholders, addressing ethical, legal, and social implications (ELSIs) related to the prevention and diagnosis of childhood melanoma. They were conducted in accordance with the Standards for Reporting Qualitative Research (SRQR) guideline [10].

Focus Group 1: The first focus group took place in person during the Melanoma Patient Pathway meeting, organized by the Romanian Melanoma Association, in Berlin on 22 March 2024 (https://www.mpneurope.org/melcaya-at-mpne, accessed on 26 December 2024). The session involved 15 participants (patients, caregivers, advocates, and a nurse) from seven different countries, all with diverse advocacy backgrounds. Recruitment was conducted among the meeting attendees, who were informed about the focus group’s purpose through an oral presentation. This presentation outlined the study’s benefits and potential risks, emphasized the voluntary nature of participation, and assured attendees of their right to withdraw at any time during or immediately after the discussion. Oral consent was obtained from each participant before the session, with confidentiality guaranteed through the anonymization of all personal data. The discussion, conducted in English, lasted 120 min and was facilitated by two moderators (PR and LS-C) and one rapporteur (CR). After presenting the results of the literature review, organized by topics and issues, the same research questions from the review (A, B, and C) were used to guide the discussion. The primary aim was to collect feedback and identify gaps in the findings. CR summarized the outcomes into key points, ensuring participant anonymity by removing any identifiable details. For this reason, no direct quote was included in the final report. Following the meeting, the collected data were reorganized based on the topics and issues effectively debated. The result of this analysis is available in the Appendix A.

Focus Group 2: The second focus group was conducted online on 19 November 2024. This session involved five participants, all researchers affiliated with the MELCAYA project. The group included a molecular geneticist, a physician epidemiologist, a data scientist, an oncologist, and an environmental epidemiologist, representing expertise from three different countries.

The recruitment process consisted of two steps: firstly, relevant participants were identified from the list of MELCAYA project researchers. They were subsequently sent an invitation letter outlining the purpose of the focus group. The letter detailed the study’s benefits and potential risks, emphasized the voluntary nature of participation, and assured invitees of their right to withdraw at any point during or after the session. Oral consent was obtained from each participant prior to the discussion.

The session, conducted in English, lasted 120 min and was facilitated by two moderators (PR and DS) and one rapporteur (CR). Similar to the first focus group, the results of the literature review, organized by topics and issues, were presented at the start. The same research questions (A, B, and C) were used to guide the discussion, with the aim of gathering feedback and identifying potential gaps. CR summarized the outcomes into key points, ensuring the anonymity of participants. No direct quotes were included in the final report. Following the meeting, the collected data were reorganized and analyzed based on the topics and issues effectively debated. The results of this process are included in the Appendix A.

## 3. Results

The three search strategies yielded a total of 205 citations, with no duplicates. After the initial screening, 184 citations were excluded as their titles and/or abstracts did not meet the eligibility criteria, and 21 were included. An additional 5 articles were identified through reference screening, resulting in a final total of 26 articles to be included. Of these, seven [11,12,13,14,15,16,17] focused on the ELSIs of melanoma prevention and diagnosis in childhood, seven [18,19,20,21,22,23,24] addressed the ELSIs of prevention and diagnosis in (ultra-)rare diseases, and twelve [17,25,26,27,28,29,30,31,32,33,34,35] explored the ELSIs of using AI for prevention and diagnosis in dermatology.

The data synthesis was conducted using a narrative approach [36], beginning with the development of the synthesis. PR and CR generated a textual summary for each study based on the data extraction forms. These summaries were then systematically organized and tabulated to align with the research questions addressed by the results.

Subsequently, the findings from the literature review were merged with those from the two focus groups. This integration was carried out through a narrative approach [7,8]. The culmination of this process is presented below. The text emphasizes the final outcomes of this integrative process, focusing on the key findings. Additionally, references to the relevant literature are provided wherever applicable to support and contextualize the results.

### 3.1. A. What Are the ELSIs Associated with Prevention and Diagnosis of Melanoma in Childhood?

#### 3.1.1. Ethical Issues

##### Lack of Adequate Public Awareness

Delayed diagnosis and insufficient prevention of melanoma in children are often due to low perceived risk and limited understanding of the disease [15]. Addressing this requires early health education programs, starting in kindergarten, to teach foundational knowledge about melanoma, sun protection, and skin monitoring. Integrating such education into school curricula fosters lifelong healthy habits and may lead to addressing any unusual or concerning changes in their skin without fear or discomfort. Additionally, tailored communication strategies using interactive and visually engaging materials are essential to effectively reach and engage younger audiences.

##### Overdiagnosis

The sixfold rise in cutaneous melanoma incidence over 40 years, despite stable mortality rates, underscores concern about overdiagnosis. Current diagnostics often fail to differentiate between harmless and aggressive melanomas, leading to overtreatment of indolent cases [13]. Overdiagnosis also diverts resources from patients in urgent need of care, raising ethical concerns about unnecessary burdens on patients [16]. While public awareness has increased diagnoses, managing these cases brings unintended consequences, including physical harm, financial strain, and psychological distress. Re-evaluating diagnostic and management strategies is essential to balance early detection benefits with the risks of overtreatment [13].

##### Underdiagnosis

Underdiagnosis is a significant challenge in CHM due to its rarity and atypical presentation compared to adult cases. Low suspicion among healthcare providers and reluctance to perform biopsies on children contribute to diagnostic delays [16]. Enhancing clinician awareness, implementing pediatric-specific guidelines, and improving education for both professionals and caregivers are crucial for early detection, which is vital for effective treatment and improved outcomes [13,15].

##### Inequality

Inequality in diagnosing and preventing CHM stems from centralized diagnostic systems limiting access to specialized care, the socioeconomic status of families influencing the diagnostic journey and outcomes, and biases in diagnosing CHM compared to adult cases, reflecting discrepancies in approaches to cutaneous melanoma [12,13,16]. Some providers hesitate to diagnose melanoma in children, even when histological parameters align with those seen in adult patients [13]. Others delay performing biopsies due to low clinical suspicion, waiting for further disease progression before intervening [16].

##### Privacy

Teledermatology, including the use of images and videos for documenting clinical history, raises ethical concerns about patient privacy [12,14]. Balancing the need for comprehensive sharing of information with patients’ preferences and priorities during diagnosis or treatment is essential.

#### 3.1.2. Social Issues

##### Beauty Ideal

For most Western societies, the societal ideal of tanned skin increases melanoma risk by promoting harmful UV exposure behaviors and undermining sun safety, particularly in children [15]. Additionally, skin diseases like melanoma carry stigma, leading to discrimination and psychological challenges that may deter children from seeking care or adopting preventive measures. Raising awareness about tanning risks and redefining beauty standards to prioritize health is crucial [15].

##### Emotional Burden

A diagnosis of CHM introduces significant psychological and emotional stress for patients and their families, compounded by financial challenges [14,16]. The rarity of melanoma often exacerbates feelings of isolation, as patients struggle to navigate the healthcare system and find a reliable physician for guidance. This challenge is further compounded by disjointed care pathways, where patients frequently encounter fragmented communication among specialists and delays in referrals. To address this, adopting patient-centric approaches throughout the diagnostic process and overall care journey is essential [11,12]. These can help alleviate emotional stress, improve patient experiences, and foster a sense of support and understanding.

##### Socioeconomic Inequality

CHM, as an ultra-rare disease, poses significant challenges in access to specialized care, as experienced professionals and healthcare centers are scarce. This scarcity disproportionately affects families from lower socioeconomic backgrounds, where financial and logistical barriers can delay diagnosis and limit access to appropriate treatment [16].

#### 3.1.3. Legal Issues

##### Privacy

Privacy issues in the context of managing rare cancers, such as CHM, manifest in multiple ways. One critical concern is obtaining informed consent when sharing patient information, including photos and videos [17]. Such documentation can be invaluable for aiding diagnosis and treatment, particularly for rare conditions, but it requires clear communication to ensure patients and their families fully understand how their data will be used and the potential implications [14].

Additionally, privacy concerns extend beyond sharing to include the secure storage of patient images and information [14,17]. Ensuring that data are protected and stored in compliance with privacy regulations is essential to safeguard confidentiality and prevent unauthorized access, which could compromise patient trust and safety [12].

##### Reimbursement

Inadequate reimbursement is a significant barrier to skin cancer screenings, discouraging primary care and advanced practice providers from incorporating these exams into routine care [15]. The time and resources required often exceed the compensation provided, creating a financial disincentive. This issue is compounded by systemic challenges such as time constraints and competing patient priorities. Revising reimbursement policies could incentivize providers to prioritize skin cancer screenings, promoting early detection and better patient outcomes.

##### Defensive Medicine

Healthcare professionals may resort to practicing defensive medicine due to concerns about the repercussions of missing a diagnosis [13,37]. This approach is primarily driven by the desire to avoid malpractice litigation and often involves excessive diagnostic testing as a precautionary measure. However, such practices may not always align with the best interests of the patient or the efficient functioning of healthcare systems, potentially leading to unnecessary procedures and resource overuse [13].

### 3.2. B. What Are the ELSIs Associated with Prevention and Diagnosis of (Ultra-)Rare Diseases?

#### 3.2.1. Ethical Issues

##### Limited Knowledge among Healthcare Professionals

An ultra-rare condition often presents atypically, differing from adult cases, making its signs and symptoms less recognizable. Physicians may have a low index of suspicion, prioritizing more common conditions, and are often reluctant to perform biopsies on children, perceiving a low risk and fearing unnecessary interventions [18]. The same applies to genetic testing [22]. To address these issues, improving healthcare professionals’ understanding of CHM is essential [21]. Comprehensive training, updated clinical guidelines, and heightened awareness are necessary to enable timely and accurate diagnosis of this rare but potentially aggressive condition.

##### Inequality

Patients with rare diseases face significant inequalities due to various factors. Geographic disparities limit access to diagnosis and treatment, with proximity to specialized centers playing a critical role [19]. Socioeconomic status further exacerbates inequality, as out-of-pocket expenses for care are often substantial [19]. The rarity of rare diseases and inconsistent epidemiological data hinder research, clinical trials, and the development of effective treatments, especially for the 50% of rare diseases patients who are children [18,19]. Diagnosis delays, often spanning years, contribute to mismanagement and a burdensome patient journey, as seen in a EURORDIS survey where 25% of patients waited 5–30 years for correct diagnoses [38]. Additionally, the absence of available treatments discourages efforts to achieve accurate diagnoses, perpetuating inequality [22,23].

##### Privacy

Privacy is a critical issue that significantly impacts the journey of patients with rare diseases [21,23]. Clinicians may choose to keep patient records private or share them only within a limited network, often to protect patient privacy or to maintain priority in publishing discoveries about causative or implicated genes. However, this practice can hinder opportunities for finding matches through broader data-sharing networks [20]. While researchers may be cautious about the risks of wider data sharing, discussions with patients are essential. Respecting the patient’s preferences helps strike a balance between avoiding unnecessary paternalism and ensuring responsible data use, fostering trust and collaboration in managing undiagnosed patients’ data.

#### 3.2.2. Social Issues

##### Isolation, Stigma, and Discrimination

Patients with rare diseases often face social isolation, stigma, and discrimination [19]. A clear diagnosis can alleviate some burdens, such as uncertainty about care pathways and access to essential healthcare, services, and subsidies [18]. However, in some cases, a diagnosis may also limit access to support if the condition is deemed untreatable [22,23]. The rise of genetic testing and data sharing has introduced new forms of discrimination, with rare diseases patients sometimes facing job denial or insurance refusal due to their condition [22,23]. Public awareness and the support of patient advocacy groups play a crucial role in reducing isolation and addressing these challenges [19].

##### Burden of the Disease

Children diagnosed with melanoma and other rare diseases, along with their caregivers, often experience significant psychological challenges, including anxiety, depression, and feelings of isolation [18]. Caregivers, in particular, bear a dual burden: managing the physical and emotional care of their child while contending with their own feelings of guilt, fear, and helplessness. To address these challenges, healthcare systems must prioritize the integration of psychosocial services into the care pathways for rare conditions [18]. Providing psychological support to parents and caregivers is crucial, as it can alleviate emotional distress and improve overall well-being.

#### 3.2.3. Legal Issues

##### Informed Consent

Critical consideration must be given to informed consent primarily in two contexts. The first pertains to incidental findings in genetic testing, which may uncover unexpected information with potential implications both for patients and for their family members [21]. This raises ethical questions about individuals’ rights to know or not know such information [20,22]. Genetic counseling services play a vital role in helping patients understand the potential outcomes of testing, including how results might influence personal health, family planning, and the well-being of relatives [21,23].

The second pertains to data sharing on matchmaking platforms: platforms that collect and compare genetic and phenotypic data to diagnose undiagnosed cases must balance the goal of broad data access with the need to protect patient confidentiality [20]. In this case, it is important that the physicians allow the patients a role in the decisions whether to be informed or not about new findings as they emerge, while still evaluating the risk/benefit ratio (including psychological harm), and always respect the right not to know [22,23].

##### Data Security

Ensuring the secure collection and storage of personal data is crucial to prevent several risks, including unauthorized access, misuse, application for unrelated purposes, and potential re-identification of individuals [20,21]. For example, individuals diagnosed with rare diseases may face discrimination, such as being denied health insurance based on their genetic test results [22,23]. This situation highlights the need to balance the confidentiality and protection of personal data with the beneficial use of test results.

##### Reimbursement

The integration of AI tools in the diagnosis of rare diseases introduces new considerations for reimbursement policies [18,19,20]. As AI technologies become more prevalent in healthcare, establishing appropriate reimbursement frameworks is essential to ensure equitable access and encourage the adoption of innovative diagnostic solutions.

### 3.3. C. What Are the ELSIs Associated with the Use of AI for Prevention and Diagnosis in Dermatology?

#### 3.3.1. Ethical Issues

##### Inaccuracy

Inaccuracy poses a significant challenge for implementing AI-driven technologies in dermatology, manifesting in various facets. The most critical issue pertains to the training of algorithms [26,27,29,32,33], which presents a vast range of challenges. Criteria for adequate images: firstly, the criteria for what constitutes an adequate image are not well studied [29]. Without standardized guidelines, the quality and consistency of images used for training AI algorithms can vary widely, affecting the accuracy of diagnoses [32]. Biased training data: since algorithms fundamentally reflect their training data [27], a biased input image dataset can directly impact their performance and generalizability [26]. If the input image dataset is biased, the algorithm may misclassify more benign nevi as malignant (false positives), potentially leading to unnecessary biopsies [27,35]. This not only causes stress to patients but also places additional demands on an already burdened healthcare system. Conversely, an increase in false negatives (lower sensitivity) could lead to inappropriate reassurance, missing malignant cases that require treatment. Population representation: existing deep learning models are predominantly trained on European or East Asian populations [26]. The lack of training on darker skin pigmentation may limit overall diagnostic accuracy. As a consequence, AI has the potential to worsen healthcare disparities, as individuals with darker skin may not receive accurate diagnoses, leading to inadequate or delayed treatment [26]. Impact on pediatric care: the inaccuracy issue could be more pronounced with pediatric data [33]. Children’s skin conditions can differ significantly from adults, and a lack of pediatric-specific training data could result in higher misclassification rates for this group. Ensuring that algorithms are trained on diverse pediatric images is crucial to improve their reliability in diagnosing skin conditions in this population [39].

##### Acceptance

Patient care extends beyond diagnosis and necessitates a holistic approach imbued with a ‘human touch’ [26,28]. While algorithms can assist in various aspects of medical practice, they cannot replace the empathetic and personal interactions between healthcare providers and patients. This limitation may lead to challenges in acceptance, as patients often seek not just clinical expertise but also emotional support and understanding from their caregivers [38].

##### Deterioration in the Doctor–Patient Relationship

AI tools, though efficient in analyzing data and aiding diagnosis, may inadvertently reduce the time and depth of interactions between healthcare providers and patients. In pediatric cases, this is particularly critical, as families often rely on empathy and reassurance from their doctor during such emotionally challenging disease journeys. The overreliance on AI tools might lead to a depersonalized approach, where patients and their families feel less supported and understood. However, it is also important to highlight the potential of AI to reduce diagnostic time, for instance, through AI-integrated videodermatoscopy. By streamlining the diagnostic process, these tools can free up valuable time for healthcare providers, allowing them to dedicate more attention to meaningful interactions with patients and their families. This perspective should be included in the balance, to emphasize that AI, when used thoughtfully, can enhance rather than undermine the doctor–patient relationship.

##### Privacy

Another significant issue is privacy, as many applications require users to agree to their data policies. The methods of collecting, using, and sharing patient data are often not transparent. This lack of clarity can lead to concerns about how personal and sensitive information is being handled. Patients may be unaware of the extent to which their data are being mined and shared, potentially compromising their privacy [17,21,30,35]. To address these concerns, it is important for app developers and healthcare providers to implement clear, transparent data policies and ensure that users are fully informed about how their data will be used and protected.

##### Responsibility

It remains unclear how responsibilities for medical malpractice will be determined if a patient is harmed due to inaccurate information provided by AI-driven technologies [27]. This ambiguity raises significant ethical questions about accountability in healthcare.

However, AI in medicine, particularly as a decision support system, may be classified as a medical device, thereby placing liability on the practitioner—much like the use of a dermoscope. In such instances, the practitioner bears responsibility unless they can demonstrate that the liability arises from flaws in the device’s design of functionality [35,40,41,42,43].

Establishing clear guidelines and regulatory frameworks is essential to address these issues and ensure that patients receive safe and reliable care. Additionally, continuous monitoring and validation of AI systems must be implemented to minimize the risk of errors and enhance the overall trust in these technologies.

##### Job Losses

There is significant debate among experts about whether AI will result in large-scale job losses [28]. Some argue that AI and automation could displace many jobs, particularly those involving routine and repetitive tasks, potentially leading to widespread unemployment in certain sectors. On the other hand, proponents believe that AI will create new opportunities and roles, driving innovation and efficiency across industries. This ongoing debate highlights the need for proactive measures, such as reskilling and upskilling programs, to prepare the workforce for the evolving job landscape shaped by AI advancements.

#### 3.3.2. Social Issues

##### Digital Literacy

There are considerable barriers to implementing AI, one of which is digital literacy. The use of technology varies based on sociodemographic factors, and more tech-savvy users are likely to be more comfortable embracing AI for tasks such as skin screening [26]. This digital divide can lead to disparities in the adoption and benefits of AI technologies. Users with higher levels of digital literacy and access to technology are better positioned to utilize these advancements, while those with limited tech skills or access may be left behind. Addressing this issue requires targeted education and training programs to improve digital literacy across diverse healthcare professionals, ensuring equitable access to the benefits of AI in healthcare. Additionally, developing user-friendly interfaces and providing support for less tech-savvy professionals can help bridge the gap and promote more widespread acceptance and utilization of AI technologies. Digital literacy, however, represents just the tip of the iceberg, as broader issues of general literacy are equally significant in shaping individuals’ ability to engage with and benefit from AI technologies.

##### Trust

Implementing AI requires trust, which is currently lacking among both dermatologists and the general population [34]. Transparency about AI algorithms, data usage, and security measures is crucial [29]. Additionally, educating dermatologists on AI integration and informing the public about its benefits and safety can foster greater acceptance and trust in AI-driven healthcare solutions.

#### 3.3.3. Legal Issues

##### Data Security

Ensuring the safety and security of data in the context of AI is fundamental and involves several key considerations. The European Union’s General Data Protection Regulation (GDPR) specifies explainability as a requirement for algorithmic decision making, a principle further reinforced by the EU AI Regulation. However, achieving this level of transparency is currently challenging and not fully attainable. Preventing third parties from accessing data during the transmission process is crucial. One potential solution is to establish a supervisory institution that oversees and ensures the integrity and security of data transmissions [25]. Differentiating between de-identifiable and identifiable data is essential for effective data regulation [33]. Properly managing these distinctions helps ensure that personal information is protected while still allowing for the beneficial use of de-identified data in AI applications.

##### Informed Consent

When choosing a method of consent in the context of AI, it is essential to consider both the feasibility of execution and the protection of patient privacy, with particular attention to vulnerable populations, such as minors. Feasibility of execution: the consent process should be straightforward and accessible to all patients, regardless of their technical literacy [17,33,35]. For minors, this involves creating age-appropriate materials and ensuring that parents or guardians are fully informed and engaged. It should be designed to facilitate easy comprehension and should not impose undue burdens on patients. Protection of patient privacy: ensuring the confidentiality and security of patient data is paramount. Consent methods should include clear explanations of how data will be protected and the measures in place to prevent unauthorized access. This includes describing how data will be anonymized or de-identified, and how patients can withdraw consent if they choose. Implementing robust informed consent processes can help build trust in AI technologies and ensure that patients’ rights and privacy are respected.

##### Reimbursement

The reimbursement mechanisms for AI technologies remain unclear [26]. Determining how these technologies will be financially supported and integrated into existing healthcare payment systems is a complex issue.

##### Responsibility

Healthcare is currently structured in such a way that responsibility and liability are carried by the provider, not the patient. This framework ensures that healthcare providers are accountable for the diagnosis, treatment, and overall care of their patients. However, the integration of AI technologies introduces complexities that challenge this traditional model [44,45].

## 4. Discussion

A thematic analysis of the most prominent themes emerging from our investigation provides valuable insights for shaping health policies [46,47,48] aimed at the prevention and diagnosis of CHM. For the purposes of this discussion, we have intentionally treated ethical, legal, and social considerations as interconnected issues, addressing them holistically to offer a comprehensive perspective.
Addressing cultural factors. Policies for the prevention and diagnosis of CHM should, first and foremost, consider societal cultural factors, such as the glorification of tanned skin as a beauty ideal and the stigma often associated with a melanoma diagnosis. Health policies must prioritize raising awareness of the risks linked to this aesthetic standard by emphasizing the dangers of excessive sun exposure and promoting healthier attitudes toward skin care. A cultural shift is needed to encourage individuals to prioritize skin protection over superficial beauty preferences. Addressing the stigma experienced by individuals with melanoma is equally important. Incorporating psychological support into health policies could provide essential counseling and resources, helping patients and their families cope with the emotional and social challenges posed by the disease.Enhancing professional training. A second cultural element to consider is the need for enhanced training among professionals in this field, particularly in light of the growing use of AI tools. The integration of AI in healthcare introduces challenges related to digital literacy, as many practitioners may lack the necessary skills to effectively use these technologies. Addressing this gap requires targeted educational initiatives to ensure that healthcare professionals are not only proficient in leveraging AI-driven tools but also capable of interpreting and applying their outputs responsibly in clinical and preventive contexts.Managing overdiagnosis and underdiagnosis. A third critical issue that health policies must address is the risk of overdiagnosis and underdiagnosis in melanoma care. Current diagnostic methods often face difficulties in distinguishing between benign and aggressive melanomas, resulting in the overtreatment of indolent cases and the undertreatment of life-threatening ones. These challenges not only divert valuable resources away from patients with urgent needs but also raise significant ethical concerns by imposing unnecessary physical, emotional, and financial burdens on individuals. Achieving a balance between the benefits of early detection and the risks of overtreatment and undertreatment necessitates a comprehensive re-evaluation of diagnostic and management strategies [37]. This issue becomes even more critical with the advent of AI tools in melanoma diagnostics, as they bring unique challenges related to accuracy and reliability [39]. A key concern lies in the training of algorithms, which depends on several factors, including the quality and representativeness of training data, the criteria for adequate imaging, and the equitable representation of diverse populations in datasets. Biases in any of these areas can compromise the performance of AI tools, potentially intensifying the problems of overdiagnosis and underdiagnosis rather than resolving them. To address these challenges, it is imperative to establish rigorous standards for the development and deployment of AI in melanoma diagnostics. This involves ensuring that training datasets are unbiased, comprehensive, and representative of the population at large. By doing so, health policies can mitigate the risks associated with both traditional diagnostic methods and AI-driven approaches, ultimately fostering a more equitable and effective framework for melanoma prevention and care.Respecting patient autonomy. A fourth crucial element to consider is what can be broadly defined as respect for individual autonomy, encompassing various aspects such as informed consent, data security, privacy protection, and the reuse of data. These challenges are particularly intensified by the growing reliance on AI in healthcare, which introduces new layers of complexity and potential vulnerabilities. Protecting data through robust security measures and ensuring compliance with privacy regulations are essential to safeguarding patient confidentiality and preventing unauthorized access. Failure to do so risks not only breaching trust but also jeopardizing patient safety, posing significant ethical and legal challenges. Central to mitigating these risks is fostering open and meaningful communication with patients. Transparent discussions about how their data will be collected, stored, and reused are pivotal to upholding their autonomy. This includes providing clear explanations about the role of AI in the decision-making process and ensuring that patients are fully aware of how AI technologies are being utilized in their care. Additionally, it is crucial to disclose that the final decision was made by the doctor after consulting AI tools, emphasizing the human oversight and professional judgment involved in the process. Such engagement allows for a careful balance between avoiding undue paternalism and ensuring the responsible management of patient data. By respecting patient preferences and involving them in decision making, health policies can cultivate trust and encourage collaborative relationships.Reducing inequalities. A fifth critical element to consider in the context of CHM is the issue of inequalities, which manifest in various forms and significantly impact patient outcomes. These disparities are driven by geographic, socioeconomic, and systemic factors, creating substantial barriers to care. Geographic disparities are particularly pronounced, as specialized care for rare conditions like CHM is often concentrated in urban or well-resourced areas. Patients in rural or underserved regions face considerable disadvantages, frequently requiring long-distance travel to access appropriate care. Socioeconomic factors further exacerbate these challenges. Individuals from lower-income backgrounds often struggle with the financial burdens associated with treatment, travel, and accommodation, which can delay or even prevent timely diagnosis and intervention. These economic barriers contribute to poorer health outcomes and increase vulnerability, especially given the lack of readily available treatments specifically tailored to CHM. Financial and logistical obstacles, such as limited insurance coverage and the high costs of extended care, further restrict access to essential medical resources. Reimbursement issues also play a pivotal role in perpetuating these inequalities. Inadequate reimbursement poses a significant barrier to skin cancer screenings, discouraging primary care providers and advanced practice clinicians from integrating these exams into routine care. This challenge is particularly acute in the case of AI technologies, where reimbursement mechanisms remain unclear, further hindering their adoption. Lastly, how the regions organize the concentration of the delivery of pediatric cancer diagnosis and treatment is also a critical issue in supporting access to specialized therapy provided by experts in each field of ultra-rare cancers. This point has been recognized by scientific societies (SIOPE) and progressively by cancer plans [40].Ethical and legal use of AI in healthcare. Lastly, there are specific considerations related to AI-based prevention and diagnostic tools that health policies must address. One key issue is responsibility. In cases where a patient is harmed due to inaccurate information provided by AI-driven technologies, it is crucial to determine how medical malpractice liability will be assigned. Establishing clear guidelines and robust regulatory frameworks is essential to address these concerns, ensuring that AI tools are implemented responsibly and that patients receive safe and reliable care. Another critical factor is the potential deterioration in the doctor–patient relationship. Overreliance on AI tools could lead to a more depersonalized approach to care, where patients and their families feel less supported, valued, and understood. This risk underscores the importance of maintaining a balance between leveraging technological advancements and preserving the human connection in healthcare.

We acknowledge several limitations in the current paper. Firstly, there is a potential risk that several relevant articles were excluded from our analysis. This limitation arises from the inherent complexity of ethical, legal, and social issues, which often overlap and can be framed in various ways depending on the research questions and perspectives adopted. As a result, the heterogeneity in framing may have led to the exclusion of studies addressing related but differently contextualized topics. Nevertheless, the findings from our review, combined with the complementary insights gathered from the focus group discussions with patients and experts, may have significantly reduced the likelihood of overlooking or omitting any critical aspect related to ELSI. Future reviews could mitigate this limitation by employing more precise definitions and selection criteria, thereby facilitating a broader and more inclusive examination of the relevant literature.

Secondly, the study’s findings are limited by the restricted number of participants involved in the focus groups, a constraint dictated by the specific tasks and scope defined within the MELCAYA project. While the qualitative insights gathered are valuable, the small sample size may limit the generalizability of the conclusions. Increasing the size and representativeness of the participant pool in future research could strengthen the robustness and applicability of the results.

Despite certain limitations, this study demonstrates notable strengths. By integrating a literature review with focus group discussions involving patients and experts, it offers a multidimensional perspective that strengthens the findings through the combination of theoretical and practical insights. Additionally, its interdisciplinary approach effectively bridges ethical, legal, and social dimensions, providing a comprehensive and holistic understanding of the topic.

## 5. Conclusions

The findings of our investigation highlight six key priorities (addressing cultural factors, such as the glorification of tanned skin; enhancing professional training for accurate diagnosis; balancing the risks of overdiagnosis and underdiagnosis; promoting patient autonomy through transparent communication; reducing inequalities to ensure equitable access to care; and making ethical and legal use of AI in healthcare) for healthcare policies aimed at improving the prevention and diagnosis of CHM, particularly in light of the development of AI-based tools. The findings emphasize the critical importance of addressing ELSI aspects when formulating health policies, particularly for rare and ultra-rare diseases, where such considerations often carry greater weight than for more prevalent conditions.

In the context of melanoma in children, adolescents, and young adults, health policies must elevate ELSI concerns to the same level of importance as other essential policy recommendations, such as expanding access to genetic testing, establishing robust disease registries, and ensuring equitable access to innovative diagnostic tools.

Together, these priorities provide a robust framework for advancing CHM prevention and diagnosis, combining the transformative potential of AI technologies with a steadfast commitment to equitable and patient-centered care.

## Data Availability

Data supporting the reported results can be found in Appendix A.

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
