# Peer review of "Policy Challenges in Ultra-Rare Cancers: Ethical, Social, and Legal Implications of Melanoma Prevention and Diagnosis in Children, Adolescents, and Young Adults"

_healthcare, 2025, doi:10.3390/healthcare13030321_

Round 1

Reviewer 1 Report

Comments and Suggestions for Authors

The article is very interesting both for the topic of CHM and for the breadth and complexity of the topics and methodology. Overall an excellent work.

The limitations certainly include the review being limited to a single dataset pubmed/medline, but disclosed and nevertheless valid if placed in the broader work of focus groups and collaborative, multidisciplinary synthesis of the evidence. The discussion nevertheless deserves bibliographical references. Especially for the ethical-legal part, more attention is paid to privacy and little to the information, involvement and role of the entitled subjects- patients and citizens - in health choices. And it is poorly referenced for the legal issues. Lines 62-65 clearly stated the aim.

Methods deserve revision in order to be complete in the text and to make references to supplementary materials only for additional and specific parts and to lighten the text. Some things are repeated and not accurate.

I would like to make some constructive observations about the work. Sorry if it is a bit lengthy. 

72 express CAYA acronym

93 after pubmed, I suggest removing the considerations on the search tool as it contains mainly clinical articles (for example articles on ELSI and AI applications could not be found on Pubmed). Moreover, a systematic review to avoid search biases, needs to be carried out in at least to main datasets, as scopus, embase …. So

96 define the years selected for inclusion

95 to 98: please revise considering thel data reported as supplementary material to avoid redundancy. I suggest clearly state in the methods the general process and in supplementary material the application for the three questions. Please also check for coherence ( it is stated that only reviews were included, so add in the exclusion criteria article not as systematic or classical review)). As inclusion and exclusion criteria, data extraction and study selection were the same, please add only in the text. Add prisma reference as dealing with a systematic review.

then line 112 clearly states which parts of the methods were reported as supplementary.

117 ethics committee approval is to be only put at the end in the specific section. please remove

oral consent: a personal answer: I am not sure that for participating in research, even only through survey and discussion, oral consent is enough. Does the ethics committee not question this issue? Thanks. 

158 remove citations as not relevant here

From 175 I suggest adding in the text the tables in the supplementary material with the articles included. Moreover, I humbly suggest developing the sentence from 174 to 175 as it is extremely relevant: authors report in this contribution the reasoned synthesis of a literature review discussed in two focus groups with citizens and experts. 

line 276 I suggest adding this reference as further discussion limits and opportunities related to liability issues. doi: 10.5826/dpc.1203a100. 

line 335 revise punctuation 

Moreover, in ultra rare cancer, another relevant issue to be disclosed regarding informed consent, is the paucity of scientific data both to empower the patient for a shared decision making and to support the diagnostic and therapeutic path.

395 also mentions this reference that deals with the limits and opportunities related to biopsy assessments. doi: 10.3390/dermatopathology10020023

411 Please also mention the role of AI in reducing diagnostic time for instance in AI integrated videodermatoscopy, then allowing more time for doctor-patient interaction, to imbalance the considerations made. 

427 Please consider and discuss briefly that AI in medicine as a decision support system could be considered as a medical device then liability is on the practitioner, as far as using a dermoscope. doi: 10.3389/fmed.2023.1337335. Otherwise the practitioner is able to demonstrate medical device projecting or functioning problems. please see some relevant article doi: 10.3390/s24113491. doi: 10.4103/ijc.IJC_399_20. doi: 10.1111/ajd.13690 in particular this one https://doi.org/10.1016/j.clindermatol.2024.06.014

3.3.3 from line 468 revise punctuation 

line 491 as previously suggested, it must distinguish the data management informed consent and the health informed consent. When applying AI for diagnosis and treatment of CHM, and in dermatology in general, the issue of information disclosure is relevant as far as authors mentioned poor digital literacy and poor comprehension in particular by the dermatologist. The issue of disclosing to the patient that that decision was undertaken by the doctor after AI use could be very relevant. Please discuss briefly also in the section 4 of the discussion

503 very relevant as HTA issue poses a responsibility burden on chief medical officer, general director and clinical engineer in the healthcare settings see 10.3233/JRS-240004 and https://doi.org/10.2196/43958.

I do not see in any part the issue of consenting by the minors, with the decisive role of the relatives, as a main issue in pediatric oncology. 

discussion to be referenced

537 and 538 see as mentioned 10.5826/dpc.1203a100. 

line 589 please revise with the ethical and legal use of ai in healthcare as liability issue is more consistent with a legal or medico-legal point of view than ethical. 

from line 600 to the end of the discussion section: limitations are properly disclosed by the methodology is robust and collaborative, so authors could balance the limitations with the real great strengths of the manuscript.

618 which are the 5 key priorities?

references 

n. 35 very relevant

n. 39 interesting, but probably could be more useful discussing also the role of research and ethics committee within the text in a international perspective.

n. 41 not relevant. If mentioned for referencing HTA methodology, please find a more consistent.

supplementary: revise according to prior considerations. Excellent the focus group reports.

Author Response

We would like to express our sincere gratitude for your thoughtful and detailed review of our manuscript. Your insightful comments and suggestions have been invaluable in improving the quality of our research and enhancing the overall clarity of the paper.

We have carefully addressed all the points you raised, and we believe that the revisions have significantly strengthened the manuscript. Please find our detailed responses to each of your comments below.

Thank you once again for your time and effort in reviewing our work. We truly appreciate your constructive feedback and hope that the revised version meets your expectations.

72 express CAYA acronym

Thank you. We removed the unnecessary acronym.

93 after pubmed, I suggest removing the considerations on the search tool as it contains mainly clinical articles (for example articles on ELSI and AI applications could not be found on Pubmed). Moreover, a systematic review to avoid search biases, needs to be carried out in at least to main datasets, as scopus, embase …. So

Thank you. We agree, and we removed the text.

96 define the years selected for inclusion

Thank you. We have defined the years.

95 to 98: please revise considering thel data reported as supplementary material to avoid redundancy. I suggest clearly state in the methods the general process and in supplementary material the application for the three questions. Please also check for coherence ( it is stated that only reviews were included, so add in the exclusion criteria article not as systematic or classical review)). As inclusion and exclusion criteria, data extraction and study selection were the same, please add only in the text. Add prisma reference as dealing with a systematic review.

Thank you very much for the valuable suggestion. We reviewed all redundancies, as suggested, by describing the general process in the main text and using the Supplementary Materials for the details. We removed the reference to PRISMA and instead referred more simply to "study selection" since the reviews we conducted are literature reviews and not systematic reviews.

then line 112 clearly states which parts of the methods were reported as supplementary.

See the previous comment

117 ethics committee approval is to be only put at the end in the specific section. please remove

Thank you. We removed the reference.

oral consent: a personal answer: I am not sure that for participating in research, even only through survey and discussion, oral consent is enough. Does the ethics committee not question this issue? Thanks.

You are absolutely right, and your request is entirely legitimate. We had extensive discussions with the ethics committee on this point, given the particular circumstances under which the focus group was conducted. In fact, it took place during a small event organized by a patient association. For the ethics committee, the key aspect was ensuring that the study's objectives were clearly explained and that participants were informed of their right to withdraw at any time. The committee approved this as an exception, to avoid overburdening the procedures.

158 remove citations as not relevant here

Citations have been removed

From 175 I suggest adding in the text the tables in the supplementary material with the articles included. Moreover, I humbly suggest developing the sentence from 174 to 175 as it is extremely relevant: authors report in this contribution the reasoned synthesis of a literature review discussed in two focus groups with citizens and experts.

Thank you so much for the suggestion. We have expanded the sentence in lines 174-175, hoping it is now clearer. We did not include the tables as well, as we believe it would make the manuscript too cumbersome.

line 276 I suggest adding this reference as further discussion limits and opportunities related to liability issues. doi: 10.5826/dpc.1203a100.

thank you for the suggestion. We added the reference

line 335 revise punctuation

Moreover, in ultra rare cancer, another relevant issue to be disclosed regarding informed consent, is the paucity of scientific data both to empower the patient for a shared decision making and to support the diagnostic and therapeutic path.

We revised the punctuation. Thank you

395 also mentions this reference that deals with the limits and opportunities related to biopsy assessments. doi: 10.3390/dermatopathology10020023

Thank you for the suggestion. The reference has been added

411 Please also mention the role of AI in reducing diagnostic time for instance in AI integrated videodermatoscopy, then allowing more time for doctor-patient interaction, to imbalance the considerations made.

Thank you for the suggestion. We have elaborated a brief text to include this point.

427 Please consider and discuss briefly that AI in medicine as a decision support system could be considered as a medical device then liability is on the practitioner, as far as using a dermoscope. doi: 10.3389/fmed.2023.1337335. Otherwise the practitioner is able to demonstrate medical device projecting or functioning problems. please see some relevant article doi: 10.3390/s24113491. doi: 10.4103/ijc.IJC_399_20. doi: 10.1111/ajd.13690 in particular this one https://doi.org/10.1016/j.clindermatol.2024.06.014

Thank you for the suggestions. We have elaborated a brief text to include these points and the related articles

3.3.3 from line 468 revise punctuation

Done!

line 491 as previously suggested, it must distinguish the data management informed consent and the health informed consent. When applying AI for diagnosis and treatment of CHM, and in dermatology in general, the issue of information disclosure is relevant as far as authors mentioned poor digital literacy and poor comprehension in particular by the dermatologist. The issue of disclosing to the patient that that decision was undertaken by the doctor after AI use could be very relevant. Please discuss briefly also in the section 4 of the discussion

Thank you for the suggestion. We have elaborated a brief text that has been incorporated into the discussion

503 very relevant as HTA issue poses a responsibility burden on chief medical officer, general director and clinical engineer in the healthcare settings see 10.3233/JRS-240004 and https://doi.org/10.2196/43958.

Excellent references! Thank you. We have taken them into consideration.

I do not see in any part the issue of consenting by the minors, with the decisive role of the relatives, as a main issue in pediatric oncology.

Thank you for the clarification. We have revised and expanded some sentences in the related paragraph on informed consent.

discussion to be referenced

537 and 538 see as mentioned 10.5826/dpc.1203a100.

Done!

line 589 please revise with the ethical and legal use of ai in healthcare as liability issue is more consistent with a legal or medico-legal point of view than ethical.

Done!

from line 600 to the end of the discussion section: limitations are properly disclosed by the methodology is robust and collaborative, so authors could balance the limitations with the real great strengths of the manuscript.

Done! Thank you

618 which are the 5 key priorities?

Thank you for the suggestion. We made the priorities explicit.

references

  1. 35 very relevant

  1. 39 interesting, but probably could be more useful discussing also the role of research and ethics committee within the text in a international perspective.

The reference has been removed

  1. 41 not relevant. If mentioned for referencing HTA methodology, please find a more consistent.

The reference has been removed

supplementary: revise according to prior considerations. Excellent the focus group reports.

Thank you. Supplementary materials have been revised

Reviewer 2 Report

Comments and Suggestions for Authors

This article offers valuable insights into the challenges of melanoma prevention and diagnosis in younger populations. Overall, it's a well-researched and engaging read. It was a pleasure to read it.

That said, there are a few areas where minor improvements could be made.

1. The introduction could provide more context for the broader challenges in rare cancer policies. As the article aims to put melanoma as an example for policy challenges in the field per se, it would be helpful to discuss what has already been outlined in the field of rare cancer policy, how melanoma compares to other rare cancers, and what EU policies are currently in place. How rare cancer differs from rare diseases. I also suggest adding these references to strengthen the introduction: (1) Kostadinov, K.; Iskrov, G.; Musurlieva, N.; Stefanov, R. An Evaluation of Rare Cancer Policies in Europe: A Survey Among Healthcare Providers. Cancers 2025, 17, 164. https://doi.org/10.3390/cancers17020164

2. Additionally, the aim of the study isn’t clearly defined. The current wording, "The following research question guided this study: what are the ethical, social, and legal implications (ELSI) related to the prevention and diagnosis of childhood melanoma?" doesn’t fully capture the study’s scope. The findings related to reimbursement (lines 357-367) are important, but they focus on treatment, which isn’t directly aligned with the main research question.

3. Finally, the flowchart, interview guides, and table with demographic characteristics should be included in the main article rather than in the attachments, for better accessibility and flow.

Overall, despite these areas for improvement, the article is strong and enjoyable to read.

Author Response

We would like to express our sincere gratitude for your thoughtful and detailed review of our manuscript. Your insightful comments and suggestions have been invaluable in improving the quality of our research and enhancing the overall clarity of the paper.

We have carefully addressed all the points you raised, and we believe that the revisions have significantly strengthened the manuscript. Please find our detailed responses to each of your comments below.

Thank you once again for your time and effort in reviewing our work. We truly appreciate your constructive feedback and hope that the revised version meets your expectations.

  1. The introduction could provide more context for the broader challenges in rare cancer policies. As the article aims to put melanoma as an example for policy challenges in the field per se, it would be helpful to discuss what has already been outlined in the field of rare cancer policy, how melanoma compares to other rare cancers, and what EU policies are currently in place. How rare cancer differs from rare diseases. I also suggest adding these references to strengthen the introduction: (1) Kostadinov, K.; Iskrov, G.; Musurlieva, N.; Stefanov, R. An Evaluation of Rare Cancer Policies in Europe: A Survey Among Healthcare Providers. Cancers 2025, 17, 164. https://doi.org/10.3390/cancers17020164

Thank you very much for the valuable suggestion. We have developed a brief paragraph on this point and included the suggested reference.

  1. Additionally, the aim of the study isn’t clearly defined. The current wording, "The following research question guided this study: what are the ethical, social, and legal implications (ELSI) related to the prevention and diagnosis of childhood melanoma?doesn’t fully capture the study’s scope. The findings related to reimbursement (lines 357-367) are important, but they focus on treatment, which isn’t directly aligned with the main research question.

Thank you very much for the valuable suggestion. We have modified the introduction to clarify the research questions. Moreover, we have removed the reference to reimbursement of treatment, as it is beyond the scope.

  1. Finally, the flowchart, interview guides, and table with demographic characteristics should be included in the main article rather than in the attachments, for better accessibility and flow.

Thank you very much for the valuable advice. You are certainly right, but to avoid overloading the text, we prefer to keep the tables in the supplementary materials.